# The Effective coalitions of Shapley value For Integrated Gradients

## Abstract

Many methods aim to explain deep neural networks (DNN) by attributing the prediction of DNN to its input features, like Integrated Gradients and Deep Shap, which both have critical baseline problems. Previous studies pursue a perfect but intractable baseline value, which is hard to find and has a very high computational cost, limiting the application range of these baseline methods. In this paper, we propose to find a set of baseline values corresponding to Shapley values which are easier to be found and have a lower computation cost. To solve computation dilemma of Shapley value, we propose *Effective Shapley value* (ES), a proportional sampling method to well simulate the ratios between the Shapley values of features and then propose *Shapley Integrated Gradients* (SIG) to combine Integrated Gradients with ES, to achieve a good balance between efficiency and effectiveness. Experiment results show that our ES method can well and stably approximate the ratios between Shapley values, and our SIG method has a much better and more accurate performance than common baseline values with similar computational costs.

## 1 Introduction

Deep Learning (DL) has exhibited significant success in various tasks, such as computer vision and reinforcement learning. Unfortunately, under the curse of transparency-performance trade-off, it's difficult to understand the intrinsic working logic of DL. Attributing the prediction of a deep network to its input features is one of the most popular methods in DL evaluation domain, such as DeepLift (Shrikumar et al. (2017)), Integrated Gradients (Sundararajan et al. (2017)) and Deep Shap (Lundberg & Lee (2017)). All these methods have a crucial problem how to choose a perfect baseline as a benchmark for input. As mentioned in Frye et al. (2020), the quality of baselines determines the quality of explanations for DL. Ren et al. (2021) considers there are two key requirements: (i) baseline values should remove all information represented by origin variable values and (ii) baseline values shouldn't bring in new/abnormal information.

Some studies provide empirical baseline values based on actual experience. Ancona et al. (2019) set baseline values as zero, Dabkowski & Gal (2017) set baseline values as mean value over many samples and usually people randomly select some samples from datasets. While other studies try to find a more reasonable baseline value. Fong & Vedaldi (2017) makes baseline values smoothed by blurring the input image with Gaussian noise. Frye et al. (2020) sets the baseline value of a pixel with surrounding pixels. Ren et al. (2021) learns baseline values corresponding to a set of features. Those methods try to approximate the perfect baseline value. However, it's difficult to find a baseline value that perfectly satisfies the two principles for various inputs in practice. For example, in the field of computer vision, zero baseline value is a common baseline value, seen as bringing no additive information. But in facial expression code task for Asian people whose eyes are black is no longer suitable for zero baseline value, owing to the black area around the eyes bringing additive information. It's ideal to use a transparent image as baseline, which is impossible to achieve in computer. Therefore, though many methods try to find a perfect baseline value, most people still use empirical baseline values based on experience, which leads to unsatisfactory and unstable results.

Instead of finding a perfect baseline value, we propose to find a set of informative baseline values, which can be found easier and have a much low computation. Shapley value (Shapley (1951)) is computed as summation of marginal difference for all coalitions and Shapley value can

accurately reflect the contributions of features. Aas et al. (2021) holds the view that Shapley value can explain the difference between prediction and global average prediction. What's more, most coalitions in Shapley value are informative and we can simply remove some features to get an informative coalition, which means can be found easier. So we propose to find a set of informative baseline values associated with Shapley value.

However, the computation of Shapley value needs to iterate over all combinations, which is exponential with respect to the number of features. For the computation dilemma of Shapley value, we propose a proportional sampling method to approximate the ratios between Shapley value and propose *Shapley Integrated Gradients* (SIG) to combine Shapley value and Integrated Gradients, achieving a good balance between efficiency and effectiveness. Integrated Gradients has a much faster computation process compared with Shapley value but we discover that though Ren et al. (2021) has pointed out that Integrated Gradients is a simulation of Aumann Shapley value alongside a special calculation path, Integrated Gradients takes a shortcut compared with calculation path of Shapley value and it will lead to an unsatisfactory and unstable explanation.

To verify the effectiveness of our sample methods, we conduct experiments on three typical tasks: human-defined function to verify the validity of our method to simulate ratio between Shapley value of players. facial expression code & image classification, to verify that our methods have better performance compared to zero baseline method or mean baseline methods;

Our contributions can be summarized as follows: (1) We discover that Integrated Gradients takes a shortcut compared with calculation path of Shapley value; (2) We propose an effective proportional sampling method Effective Shapley value to approximate the ratios between Shapley values and design experiments to verify the effectiveness and preciseness of our methods; (3) We propose Shapley Integrated Gradients which combines Integrated Gradients with Effective Shapley value and achieve a balance between efficiency and effectiveness.

## 2 RELATED WORK

Most previous studies focused on finding a perfect baseline value, while our proposed method try to find a set of baseline values that are informative and easier to obtain. In our proposed methods, the selection of baseline values is based on the calculation path of Shapley value.

**Shapely value**. Shapley value (Shapley (1951) was proposed to distribute contributions to players, assuming that they are collaborating. It's the only distribution with desirable properties, linearity, nullity, symmetry, and efficiency axioms. Aumann & Shapley (2015) extended the concept of Shapley value to infinite game. Some previous studies used Shapley value for model explanation. Lundberg & Lee (2017) proposed Shapley Additive exPlanations(SHAP), a model explanation method with Shapley value. The SHAP regards the feature as a player in game, regards model as utility function, and uses chain rule to reduce computational complexity. Based on SHAP, Lundberg et al. (2018) continued to propose TreeSHAP, a method for tree model reducing complexity from $O(TL2^{\hat{M}})$ to $O(TLD\hat{2})$. It's worth noting that there is also a baseline problem in SHAP. To simplify computation, Ghorbani & Zou (2019) used Monte Carlo Sampling and gradient-based methods to efficiently estimate data Shapley values. Ancona et al. (2019) sampled from distribution of coalition size of k and then average all these marginal contributions as an approximation of Shapley value. Ghorbani & Zou (2020) proposed a new multi-armed bandit algorithm to explain Neuron's Shapley value.

**Integrated Gradients**. Integrated Gradients was proposed by Sundararajan et al. (2017) to combine implementation invariance of gradients along with the sensitivity of techniques, which also needs a crucial baseline value. Merrick & Taly (2020); Kumar et al. (2020); Binder et al. (2016); Shrikumar et al. (2017) provided their experiential guidance of selecting baseline values, without providing any theoretical guidance. And Chen et al. (2021) regarded baseline values as background distribution, which is similar to our view.

## 3 PRELIMINARIES

**Shapley value**. Let us consider a game with $n$ players and $F = S|S \subseteq 2^n$ means all subsets of players. Game will return a reward corresponding to the coalition $S$ through utility function $v$.

For coalition $S$ and player $x_i$ where $x_i \notin S$, the marginal contribution of player $x_i$ with coalition $S$ is $v(S \cup x_i) - v(S)$. Shapley value of player $x_i$ is the summation of all coalition $S \in F/i$ with probability of selecting coalition, as follows:

$$f(x_i) = \sum_{S \in F/\{i\}} \frac{|S|!(|F| - |S| - 1)!}{|F|!} (v(S \cup x_i) - v(S)). \tag{1}$$

**Aumann Shapley value**. With the expansion to infinite game, Aumann Shapley value was proposed with definition that $ds$ represents infinitely small player in game, $I$ represents the complete set of players and $tI$ is a perfect sampling with scale $t$. Aumann Shapley value can be written as follows:

$$f(ds) = \int_0^1 v(tI + ds) - v(tI)dt. \tag{2}$$

**Integrated Gradients**. Integrated Gradients was proposed to attribute the prediction of a deep network to its input features. It integrates the gradient of the prediction with respect to input features along a straight path between input $x$ and baseline $x'$. Integrated Gradients for model $F$ can be written as follows:

$$IntegratedGrad_i(s) = (x_i - x'_i) \times \int_{\alpha=0}^1 \frac{\phi F(x' + \alpha(x - x'))}{\alpha x_i} d\alpha. \tag{3}$$

## 4 OUR METHOD

### 4.1 PROBLEM OF INTEGRATED GRADIENTS

#### 4.1.1 THE CONNECTION BETWEEN INTEGRATED GRADIENTS AND SHAPLEY VALUE

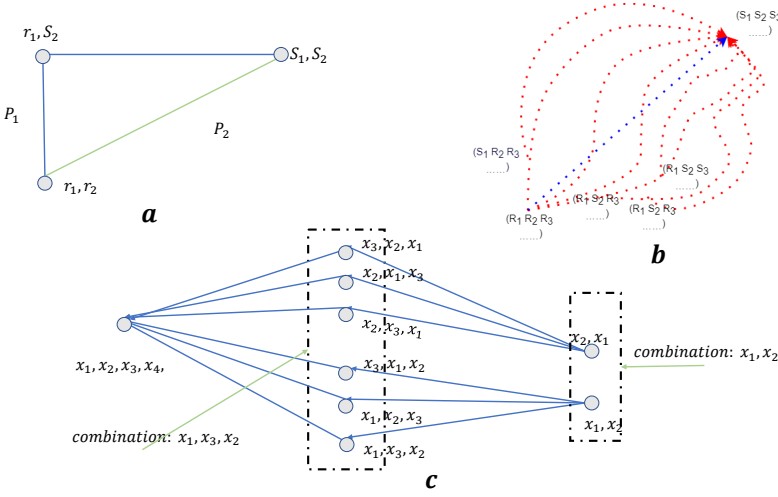

Figure 1: (a) The calculation paths of Integrated Gradients and Shapley value in two features input. Path $P_1$ is the calculation path of Shapley value while $P_2$ is Integrated Gradients; (b) Orange path is the calculation path of Shapley value while blue path is the calculation path of Integrated Gradients; (c) The calculation path of permutations between $(x_1, x_2)$ and $(x_1, x_2, x_3)$.

We suppose that there are two dimension features as shown in Figure 1(a), baseline sample $x' = (r_1, r_2)$ and input sample $x = (s_1, s_2)$. Integrated Gradients computes feature's contribution alongside path $P_2$ according to Sundararajan et al. (2017). Baseline sample $x'$ is considered as not containing any information and all points in path $P_2$ can be regarded as joining the calculation of contribution for each feature. We make the hypothesis that baseline sample $x'$ is considered as background in game and all points alongside $P_2$ are considered as complete set $I$ in game. We can find that Integrated Gradients is the same to Aumann Shapley value under this hypothesis. The same

can be extended to $n$ dimensions. Formally, Integrated Gradients is an approximation of Aumann Shapley value along Path $P_2$. The Proof of theorem 1 is shown in Appendix A.1.

**Theorem 1**: Integrated Gradients is an approximation of Aumann Shapley value alongside path that starts from baseline value $(r_1, r_2, ..., r_n)$ and directly connect with end point $(S_1, S_2, ..., S_n)$.

### 4.1.2 THE LIMITS OF INTEGRATED GRADIENTS

Integrated Gradients simulates Atumann Shapley value among points in $P_2$. However, let's consider Shapley value. In Figure 1(a), we regard baseline sample $x'$ as $\varnothing$ in Shapley value and $S_1$ as a player and points $(r_1, r_2), (r_1, S_2), (S_1, S_2)$ as coalition. Shapley value of player $S_2$ is defined as:

$$f(s_2) = \frac{1}{2}(v(r_1, s_2) - v(r1, r_2)) + \frac{1}{2}(v(s_1, s_2) - v(s_1, r_2)). \tag{4}$$

When computing Shapley value, we will walk through points $(r_1, r_2), (r_1, S_2), (S_1, S_2)$. We call points to be walked through when computing Shapley value *calculation path of Shapley value*. Path $P_2$ is a shortcut for Shapley value, conflicting with path $P_1$.

Now, let's take a closer look at computation on coalitions. As a matter of fact, marginal difference between $S \cup x_i$ and $S$ is made up marginal difference between permutations in $S \cup x_i$ and permutations in $S$. Points in combinations is permutations of combinations and factor $\frac{|S|!(|F|-|S|-1)!}{|F|!}$ can be seen as proportion of connections/lines between $S \cup x_i$ and $S$ for all connections/points since permutations wouldn't affect utility value due to nature of Shapley value. In Figure 1(c), we simulate computation between $(x_1, x_2)$ and $(x_1, x_2, x_3)$. Factor $\frac{2*2}{24} = \frac{1}{6}$ means lines between $(x_1, x_2)$ and $(x_1, x_2, x_3)$ take a proportion of $\frac{1}{6}$ for all lines in the computation of Shapley value for $x_3$. For simplification, we hide the computation process of permutations, just show computation of combinations and set weight to combination according to the factor.

In conclusion, the calculation path of Shapley value is the weighted points walked through when calculating Shapley value. As shown in Figure 1(b), the calculation paths of Shapley value are presented with orange dotted lines, and the blue dotted curve represents the calculation path of Integrated Gradients.

### 4.2 EFFECTIVE SHAPLEY VALUE

For Shapley value of player $x_i$, if we collect all coalitions without $x_i$, we can get accurate Shapley value with the formula of Shapley value, which will bring a large computation. We notice that not all coalitions contribute to Shapley value for player $x_i$. Coalitions that satisfies $v(S \cup x_i) - v(S) = 0$ actually don't influence Shapley value for player $x_i$. We call coalitions satisfy that $v(S \cup x_i) - v(S) \neq 0$ *effective coalitions for player $x_i$*. So we just need to collect effective samples for player $x_i$ and then can get accurate Shapley value. However, it still needs to iterate over all effective coalitions, still having an extensive computational cost. Considering that in actual cases we are usually concerned about which player/feature has more contribution than others, not the exact Shapley value of each player. We transform our goal from accurate Shapley value to ratio between Shapley value of each player. We can collect proportional coalitions from effective coalitions.

When we classify effective coalitions according to number of players in the coalition, in formula $S \in N$ and $N$ is effective coalitions for player $x_i$, we can easily simulate ratio between Shapley value of players through proportional sampling. Let's consider the influence of factor $\frac{|S|!(|F|-|S|-1)!}{|F|!}$. In the analysis of section 3.1, factor represents proportion of the coalition's permutations in all permutations. In theory, we can directly multiply factor with marginal difference of coalition $S$, which will be slow if we iterate one by one. For set $d_{jk}$ that satisfies $v(S \cup x_i) - v(S) = j \& |S| = k, S \in d_{jk}$, its contribution to Shapley value consists of factor $\frac{|S|!(|F|-|S|-1)!}{|F|!}$ and number of coalitions in $d_{jk}$. In order to seep up calculation, we hope to reflect influence of factors through numbers of sampling coalitions. Concretely, we set $m_j$ as coalition set that satisfies $v(S \cup x_i) - v(S) = j$, $normalization$ as normalization function and $|T|$ as sample number for player $x_i$. The sampling number $C_{jk}$ of set $d_{jk}$ is

$$C_{jk} = normalization(\frac{m_j}{\sum m_j} \frac{|k|!||F| - k - 1||!}{|F|!})|T|. \tag{5}$$

The overall algorithm of our *Effective Shapley value* (ES) is shown in algorithm 1. We first randomly select partial coalitions $A$ and then classify them by marginal difference $v(S \cup x_i) - v(S), S \in A$, recorded as $M = (m_1, ..., m_j, ...)$. For $m_j$, we then classify by players of coalition $|S|, S \in m_j$, recorded as $D = (d_{1j}, ..., d_{jk}, ..,)$. Lastly according to equation 1, sample from $d_{jk}$. Theorem 2 gives the probability of correct results. The Proof of Theorem 2 is shown in Appendix A.2.

**Theorem 2**: Make $n_j$ considered to be all combinations without player $j$, $e_j$ considered to be effective coalitions in $n_j$, $b_j$ considered to be randomly selected coalitions in $n_j$, $a_j$ considered to be effective coalitions in $b_j$, $a_{min} = \min(a_j) \in [1, A]$ that the range of minimum of $a_j$. The correct probability of proportional sampling for Shapley value of player $j$ is: $P = \sum_{k=1}^{A} \prod_j C_{|e_j||a_j|} C_{|n_j|-|e_j|}^{|b_j|-|a_j|}$.

---

**Algorithm 1** Effective Shapley value

> **INPUT**: player set $n$, utility function and constant $T$
> **OUTPUT**: baseline coalitions for each player
> Randomly select samples from subsets grouped by length
> **for** $x_i$ in player set $n$ **do**
>    $A \leftarrow$ select effective sample set of player $x_i$
>    classify S into $m_i$ according to $v(S \cup x_i) - v(S) = i \quad S \in A$
>    **for** $m_j$ in $M$ **do**
>       Classify S according to $|S| = k \quad S \in m_j$
>    **end for**
>    Get $D = (d_{j1}, ..., d_{jk})$
>    Randomly sample according to equation 1
> **end for**

---

**Algorithm 2** Shapley Integrated Gradients

> **INPUT**: player set $N$, utility function and constant $T$
> **OUTPUT**: baseline samples for each player
> Set $m$ players as a single player and get new player set $M = (m_1, ..., m_n)$
> **for** $m_i$ in player set $M$ **do**
>    Sample effective coalitions $D_i$ for $m_i$ according to algorithm 1
>    Set $D_i$ as baseline values and set coalition with all players as input
>    Compute Integrated Gradients over these baseline coalitions
> **end for**
> Average all values of Integrated Gradients as each player's $n \in N$ contribution

---

### 4.3 SHAPLEY INTEGRATED GRADIENTS

As a matter of fact, for coalitions sampled by algorithm 1, there is no need to apply Integrated Gradients, just directly computing marginal difference of coalitions and average it as contribution of player $x_i$. However, when there are many players, like pixels in image, usually meaning 244*244 players, it's almost impossible to compute each player's sampled coalitions. So we set part of image as player like 80*80 area as a player and apply Integrated Gradients to compute each player's contribution. Concretely, We set these sampled coalitions as a set of baseline values and set input as origin input. Then we average contributions computed by Integrated Gradients and get each feature's contribution. From point of calculation path, this method starts from some points in calculation path of Shapley value and ends with input. It's worth pointing out that as the area decreases, the result will get closer to ratio between Shapley value. When the area decreases to 1, result will be close to ratio between Shapley value. The problem become that we set sampled coalitions as baseline values and coalition containing all players as input. Under the guarantee of theorem 3, The Proof of theorem 3 shown in Appendix A.3, result will get close to ratio of Shapley value. Shapley value verify experiment will prove it.

The overall algorithm *Shapley Integrated Gradients*(SIG) is shown in algorithm 2. We first set players coalition as a single player, like 80*80 area as a player in image. Then we select partial

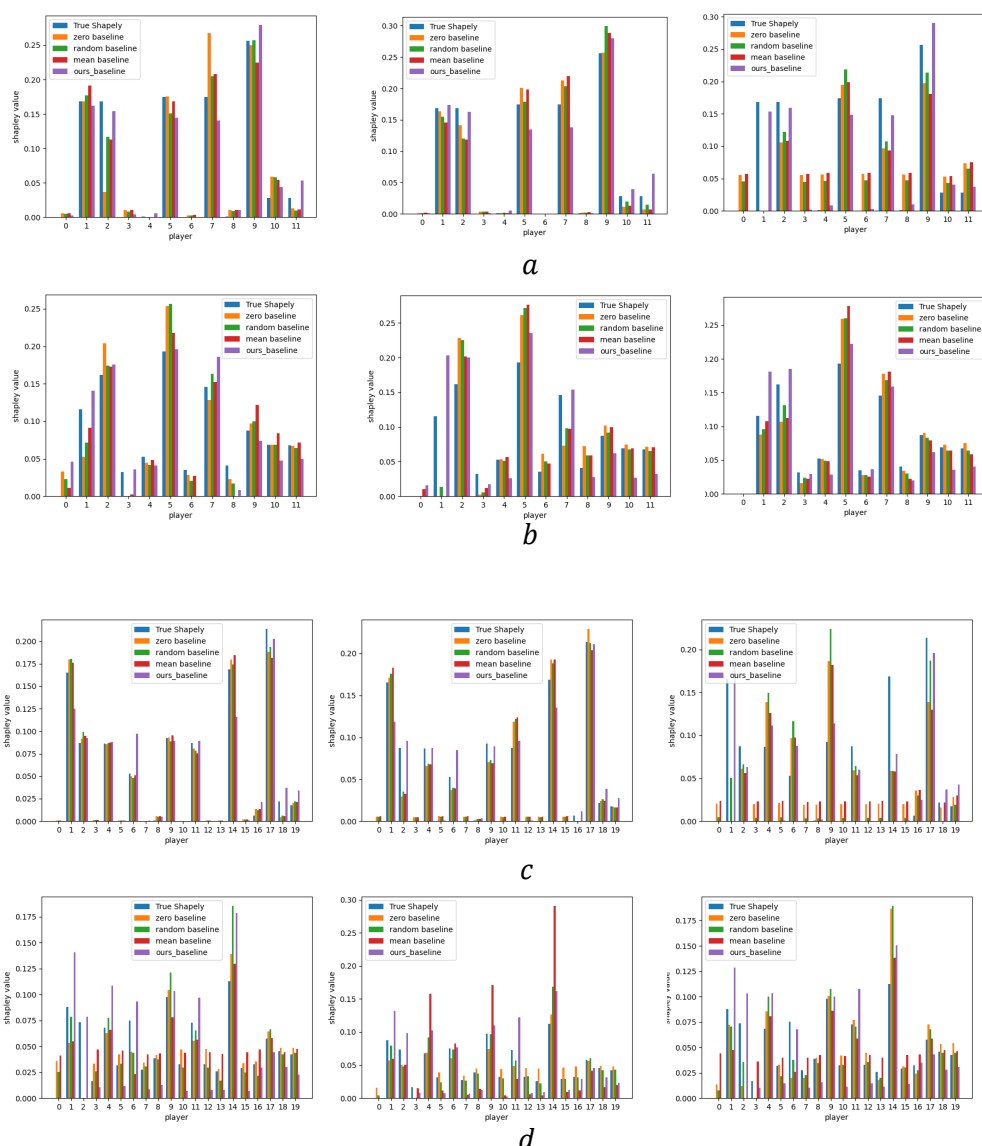

Figure 2: (a) represents human defined utility function (1); (b) represents human defined function (2); (c) represents represents human defined utility function (3); (d) represents human defined utility function (4). Each row represents an independent experiment.

effective coalitions of these *players* according to algorithm 1. We set these selected coalitions as baseline values and set coalitions with all players as input, like origin image to compute Integrated Gradients. Finally, we average these values as each player's contribution.

**Theorem 3**: Integrate Gradients between effective coalition $S$ and coalition $S \cup x_i$ for player $x_i$ can exactly equal marginal difference in Shapley value under the hypothesis that in utility function $v$ coalition $S$ directly connect coalition $S \cup x_i$.

While we can also set zero baseline value as the baseline value and set our sampled coalitions as input. Interestingly, in ResNet(He et al. (2016)), there isn't much difference between these two methods as shown in Appendix B.1., which is opposite in DLN. It's worth pointing out that our SIG computation process is much faster due to the batch computation of gradients.

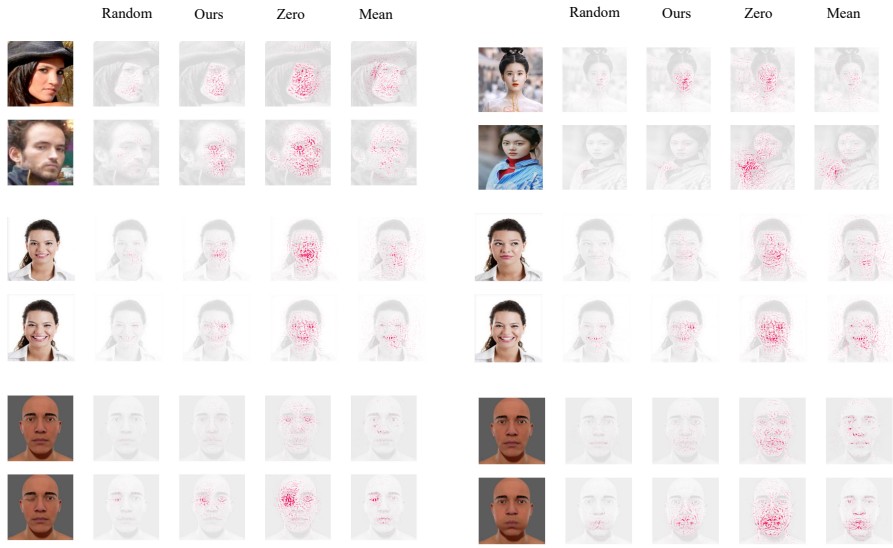

Figure 3: Integrated Gradients value produced with different baseline values on facial expression coding task. From results, we can obviously find that our SIG method can concisely and accurately locate relevant features.

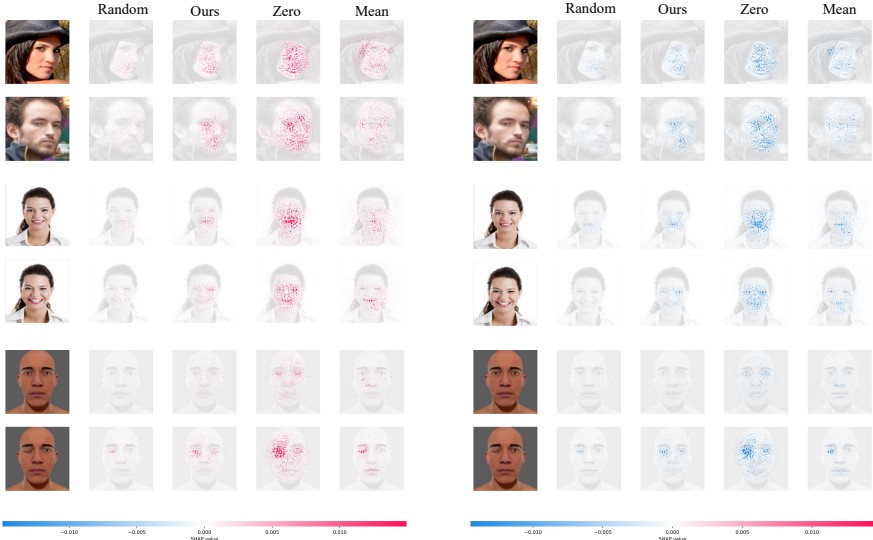

Figure 4: Comparison of distribution over positive values and negative values. The distribution of positive values is similar to negative values and our SIG achieves a better result on both distribution of positive and negative values.

## 5 EXPERIMENT

To prove the validity of EG, in other words, the simulation of ratio between Shapley value of players, we experiment with our ES method on human defined utility function, which has a few features and much less coalitions than actual application. To verify the validity of SIG, we choose Facial expression coding & Image classification tasks, which have 244*244 features.

**Env**: (1) Human defined utility function, we define a utility function $v$ with $n$ players; (2) DLN(Zhang et al. (2021)) model for Facial expression coding: a model to learn compact and

identity-invariant facial expression embedding by explicitly disentangling the identity attribute. The model takes two pictures as input and outputs distance between two pictures; (3) ResNet a classic computer vision model to classify image.

**Baseline Sample**. We choose three baseline methods that are widely used for various tasks: (1)Mean baseline values. The baseline value of each input variable is set to mean value of this variable over all samples; (2)Zero baseline values, baseline values of all input variables are set to zero;(3)Random baseline samples, randomly selected baseline samples from dataset.

**Shapley value verify**. To verify the generality of EG, we define four utility function: (1) 12 players; (2) 12 players with different utility of coalitions; (3) 20 players; (4) 20 players with same utility function to (2). We assume that utility function will have a much high computation cost corresponding to actual cases so we use DNN model to simulate utility function. The DNN model can perfectly replace utility function with partial data, with results shown in Appendix B.2. In (1)&(2), we choose 50% coalitions as train dataset. In (3)&(4) we choose 40% coalitions as train dataset. We set coalitions with all players as input and set our sampled coalitions as baseline values. Considering that we randomly sample partial coalitions firstly in ES, we conduct three independent experiments to avoid the influence of randomness.

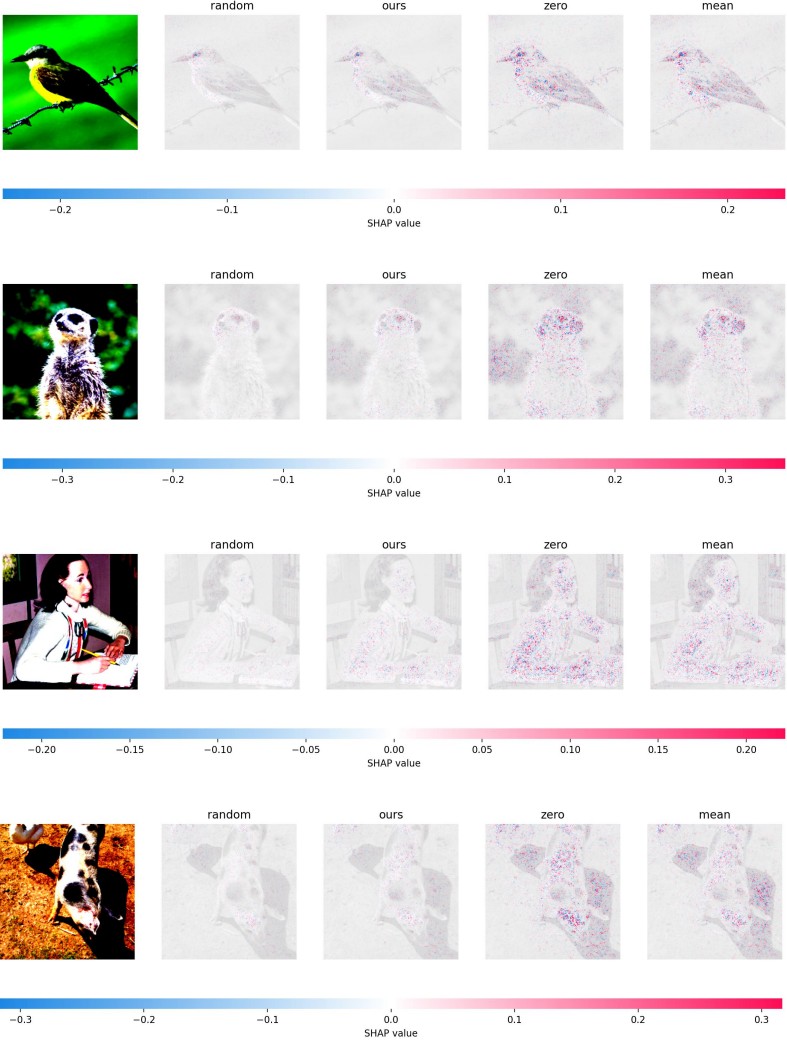

Figure 5: Experiments on Resnet and ImageNet dataset with different baseline values. We can obviously find out that our SIG method is similar with random baseline which is the common baseline method but more concerned about areas in zero and mean baseline method.

As shown in Figure 2, we can discover that in some cases, ES achieves similar results compared to other three baselines, but in other cases, the results of other three baselines are quite disastrous while ES shows a relatively accurate results. It shows that ES not only performs better than other three baselines but is more stable than other three baselines.

**Baseline Verify**. In computer vision tasks, if we treat each pixel as a player, there would be 244*244 players, which is acceptable computation. In order to reduce computation and compute each pixel's contribution, we treat 80*80 pixel area as a player and apply Integrated Gradients to compute each player's contribution. We set our sampled baseline coalitions as baseline values and set coalition with all players as input.

As shown in Figure 3, we experiment with our SIG method in the facial expression codeing task. We select positive points that $values > 0$ which will reduce the distance between two pictures in our views because negative points that $values < 0$ which have the opposite effect. And we discover that negative points have a similar distribution with positive points as Figure 4 shows. If we mix positive and negative values, this will lead to poor interpretability.

Compared with other three baseline methods, our SIG results are more concentrated near human facial organs, such as eyes, lips, etc, more in line with human intuition both in negative and positive values. Although Zero and mean baseline methods can also identify human facial organs, they will also pay attention to other areas of human face, in some cases even focusing on areas beyond the scope of human faces. More excitedly, the time cost by our SIG method is close to other baseline methods. More details are in Appendix B1.

And as shown in Figure 5, we experiment with our SIG method in the image classification tasks based on ResNet. We set almost same configuration to facial expression coding task. The performance of our SIG method is similar to random baseline, but more focused on areas that zero baseline and mean baseline pay attention to while random baseline ignores.

## 6    CONCLUSION

In this paper, we discover that Integrated Gradients takes a shortcut compared with Shapley value, which leads to unsatisfactory and unstable results. We propose Effective Shapley to find out a set of coalitions, which corresponds to calculation of Shapley value. In order to reduce computation and get a more applicable DNN explainer, we propose Shapley Integrated Gradients which combines Integrated Gradients with Effective Shapley. Experiments show that our Effective Shapley value method can well approximate the ratios between the Shapley values of players and our Shapley Integrated Gradients method can achieve better and more stable performance than Integrated Gradients.

For future work, we will explore the connections among features, set related features as a single player , and try to find a more reasonable coalition set. What's more, we will improve the random sampling method and get closer to Shapley value and apply Effective Shapley value to complex applications.

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

# A PROOF OF THEOREM

## A.1 PROOF OF THEOREM 1

First, we rewrite Aumann Shapley value's formula into gradient form as follows:

$$
\begin{aligned}
f(ds) &= \int_0^1 v(tI + ds) - v(tI)dt \\
&= \int_0^1 \frac{v(tI + ds) - v(tI)}{ds} ds dt \\
&= ds \int_0^1 \frac{\varphi\, v(tI)}{\varphi s} dt.
\end{aligned}
$$

Considered that coalition usually plays game on the basis of game's background playground. The utility of coalition can be regarded as the utility of coalition with background, $v(tI) = v(tI + \varnothing)$. The Aumann Shapley can be written as:$f(ds) = ds \int_0^1 \frac{\varphi\, v(tI+\varnothing)}{\varphi s} dt$. Integrating from the empty set/background of game $\varnothing$ to the complete set I, we can get the Aumann shapley value of the complete set:

$$
f(I) = \int_\varnothing^I \int_0^1 \frac{\varphi\, v(tI + \varnothing)}{\varphi s} dt ds.
$$

.

Since in Integrated Gradients baseline sample $x'$ is considered of containing nearly zero information, we set baseline sample $x'$ as the background of game. As complete set $I$ means all players that join the game, points alongside $P_2$ participate in contribution allocation problem in Integrated Gradients. We make all points alongside $P_2$ as $I$.

$$
\begin{aligned}
f(I) &= \int_\varnothing^I \int_0^1 \frac{\varphi\, v(t(x - x') + x')}{\varphi s} \\
&= (x - x') \int_0^1 \frac{\varphi\, v(t(x - x') + x')}{\varphi s} dt ds.
\end{aligned}
$$

Noticing that $s$ is a vector, for each feature, we can get Integrated Gradients of feature $x_i$:

$$
f(s_i) = (x_i - x_i') \int_0^1 \frac{\varphi\, v(t(x - x') + x')}{\varphi x_i} dt ds.
$$

Proof over.

## A.2 THE PROOF OF THEOREM 2

We can model the problem as independent samples. So For $n_j$, the probability of selecting k effective sample

$$
P_k = C_{|e_i|}^k C_{|n_i| - |e_i|}^{|b_i| - k}.
$$

For perfectly proportional sampling, we limit the proportion without constraint of the number of $a_{min}$

$$
a_1 : a_2 : ... : a_N = e_1 : e_2 :, .., : e_N.
$$

The probability of sampling correct number of samples is

$$
P = \sum_{k=1}^A \prod_j C_{|e_j|}^{|a_j|} C_{|n_j| - |e_j|}^{|b_j| - |a_j|}.
$$

### A.3 THE PROOF OF THEOREM 3

The hypothesis means gradients between $S$ and $S \cup x_i$ are a constant

$$\frac{\varphi v(t(x - x') + x')}{\varphi s} = \frac{v(I) - v(\varnothing)}{I}, t \in [0, 1]$$

.

Following equation, we prove Theorem 3:

$$\begin{aligned} f(I) &= I \times \frac{v(I) - v(\varnothing)}{I} \\ &= v(I) - v(\varnothing) = v(I + \varnothing) - v(\varnothing) \\ &= v(S \cup x_i) - v(S). \end{aligned}$$

## B  MORE EXPERIMENTAL RESULTS AND DETAILS

### B.1 DISCUSSION ABOUT BASELINE VALUES IN COMPUTER VISION

For ResNet, we choose origin image as input of Integrated Gradients and make coalitions selected by our sample method as baseline values, noted proposal 1. To test validation of our sampled coalitions, we choose zero baseline values as baseline values and choose our sampled coalitions as inputs and average contributions, noted proposal 2. To our surprise, there isn't much difference between the two methods, as shown. In consideration of computational efficiency we choose coalitions selected by our method as baseline values and choose origin image as input, since PyTorch performs optimization of gradient calculation and it's faster to set origin image as input. From left to right, it's proposal 2, proposal 1, proposal 1 + proposal 2, zero baseline, mean baseline, and random in each picture.

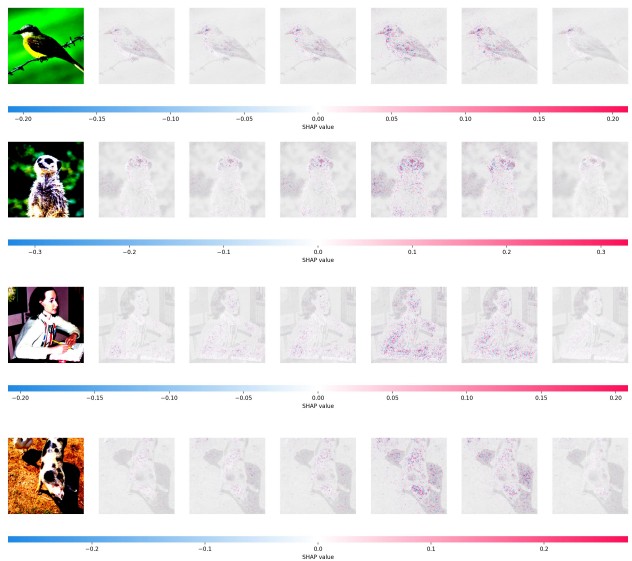

Figure 6: The effect of different configuration for ResNet on ImageNet.

We set the same proposals to ResNet. Results are shown in Figure 7. From left to right, it's proposal 2, proposal 1, zero baseline, mean baseline and random in each picture.

From the result, we can obviously find that our method has a similar explanation to proposal 2 in ResNet, but our method has a better explanation for facial expression task. Moreover, our method is much faster than proposal 2.

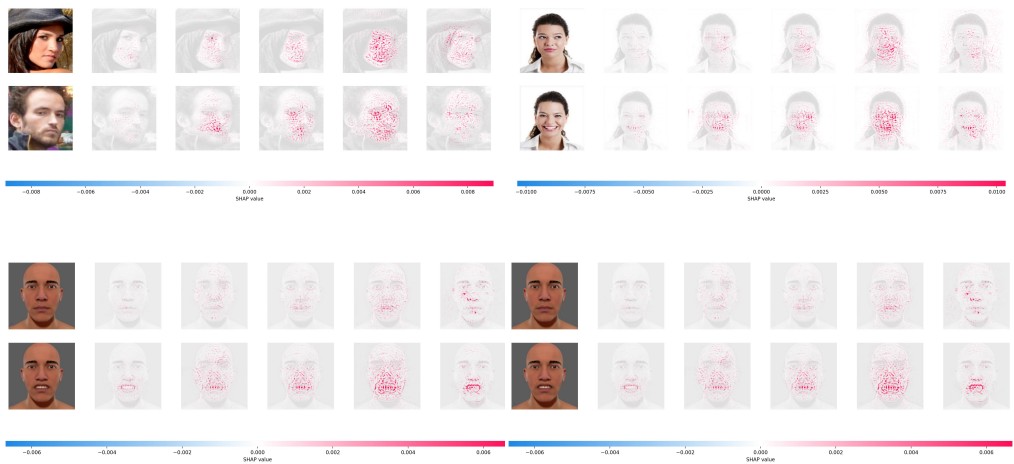

Figure 7: The effect of different configurations for DLN on facial expression task.

## B.2 DDDITIVE EXPERIMENT FOR HUMAN DEFINED UTILITY FUNCTION.

### B.2.1 VERIFY OF DEEP NEURAL NETWORKS

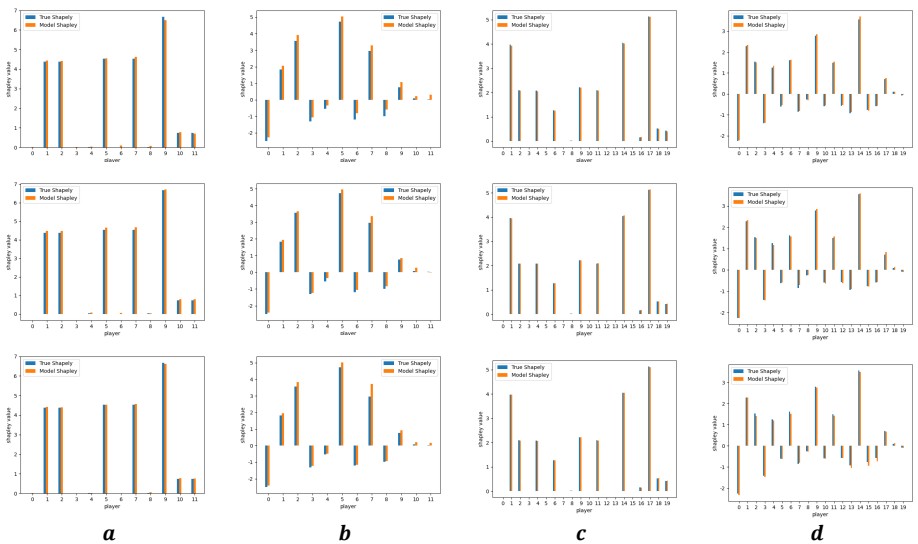

Figure 8: The effect of our trained deep neural network.

We use trained deep neural networks to replace utility function in the computation of Shapley value, as shown. Each row represents an independent experiment. It can be obviously found that our trained deep neural network can perfectly replace utility function, leading to almost same Shapley value. $a$ represents Env (1) ; $b$ represents Env (2); $c$ represents Env (3); $d$ represents Env(4).

### B.2.2 DIFFERENCE BETWEEN INTEGRATED GRADIENTS AND MARGINAL DIFFERENCE

We accumulate marginal difference and Integrated Gradients between $S \cup x_i$ and $S$, where $S \in F/i$ and $x_i$ represents $i$ player. From the result, we can view that $S \cup x_i$ linearly connect $S$

in deep neural network and the result also proves that our correctness of Theorem 2. $a$ represents Env (1) ; $b$ represents Env (2); $c$ represents Env (3); $d$ represents Env(4). Each row represents an independent experiment.

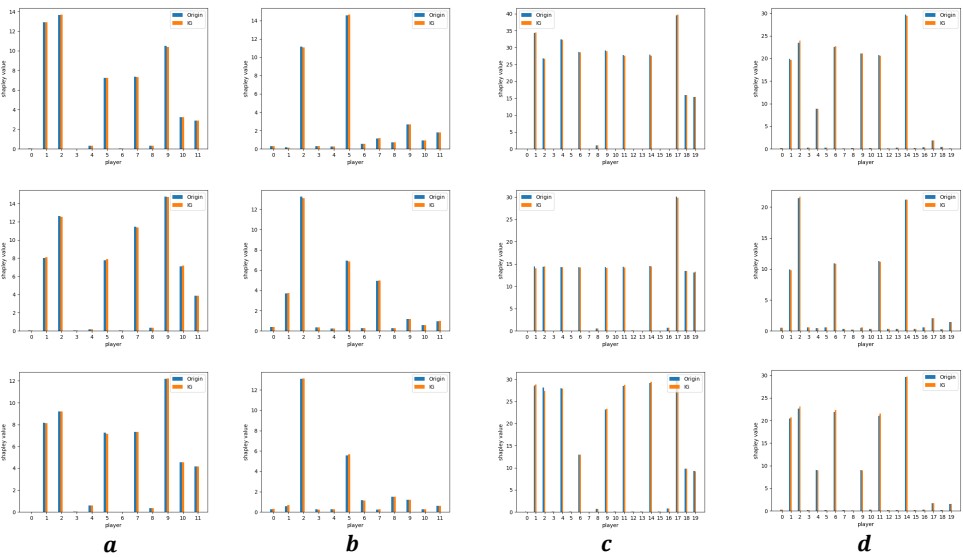

Figure 9: The difference between Integrated Gradients and marginal difference based on sampled coalitions.

