# OpenReview forum: "The Effective coalitions of Shapley value For Integrated Gradients"
_ICLR.cc/2023/Conference — Submitted to ICLR 2023_

### Official Review · Reviewer_cr3F · 2022-10-24

**Confidence:** 3
**Correctness:** 2
**Technical Novelty And Significance:** 2
**Empirical Novelty And Significance:** 2
**Recommendation:** 3

**Clarity, Quality, Novelty And Reproducibility:**

As explained above, I think that the quality and clarity of the paper require a major revision.
It is also the case that the writing of the paper makes it difficult for me to judge the novelty of the approach.


**Strength And Weaknesses:**

In my view, the biggest weakness of the paper is the clarity which requires a major revision.

1. The contribution of the paper is not clear. In the 3rd paragraph of the introduction, the paper states that its main contribution is to "propose to find a set of informative baseline values". However, this paragraph discusses the Shapley value as a measure of the feature contribution without discussing how baseline values will be chosen in this paper (as opposed to existing baselines: e.g., zero baseline values).

2. The Integrated Gradient approach is motivated by 2 axioms including the implementation invariance. Thus, it is not convincing that this work criticizes Integrated Gradients by proposing an approach (Shapley value) that not only does not satisfy the implementation invariance but also requires more computation than the Integrated Gradients.

3. Furthermore, while Section 4.1.2 is about the limits of Integrated Gradients, it only discusses the Shapley value. I do not understand the limits of Integrated gradients after reading this section.

4. The paper criticizes that "Integrated Gradients takes a shortcut compared with calculation path of Shapley value and it will lead to an unsatisfactory and unstable explanation". However, it uses the Integrated Gradients in Algorithm 2, instead of computing the marginal contribution of coalitions. Thus, why the "unsatisfactory and unstable" of the Integrated Gradients is prevented in this work?

5. After section 4.2, I still do not understand how to estimate the ratio between Shapley values even though it is the main objective of this section. In particular, how does the proposed method "transform the goal from accurate Shapley value to ratio between Shapley value"?

6. In Theorem 2, P is not in the range of [0,1] so why is it even a probability?

7. In the first paragraph of Section 4.3, it discourages the computation of the marginal difference of sampled coalitions by Algorithm 1 because it is impossible for a large number of players. However, algorithm 1 requires computing these marginal differences when it classifies S into m_i. Does it mean that algorithm 2 is very computationally intensive since it performs computing these marginal differences when invoking algorithm 1? Moreover, this work should include analyses of the computation complexities of the two algorithms.

8. The paper also mentions the 2 key requirements of baselines in the first paragraph of the introduction, but it never says how the proposed baseline values satisfy (or do not satisfy) these 2 requirements.

9. Why does the factor for d_jk not change even though we have an additional requirement that v(S cup x_i) - v(S) = j? According to (1), the factor only depends on the size |S|.

10. In Theorem 2, the explanation is confusing: is a_j a set of coalitions or just a coalition ("a_j considered to be effective coalitions in bj")? What does it mean by min(a_j)?

11. Algorithm 2 is confusing because it says the output is baseline samples for each player (in the 2nd line). However, the last line of the algorithm outputs the average of all values of IG.

There are other minor issues.

12. In the line below equation (4), where is the point (s1,r2)?

13. Typo in C_{|ej|}^{|aj|} in Theorem 2.

14. "seep up" -> "speed up"

15. D = (d_1j, ..., d_jk, ...) is confusing due to the notation j.

16. What is \phi in equation (3)?

17. Figure 1.b, there is not any orange color.

18. In the 2nd line of the paragraph after Figure 1, x=(s1,s2) should be x=(S1,S2).


**Summary Of The Paper:**

This work proposes a proportional sampling method to estimate the ratios between Shapley values by introducing effective coalitions. Then, it combines this method and the existing Integrated Gradients to propose the Shapley Integrated Gradients that can explain a deep neural network (DNN) prediction by attributing the prediction to its input features.


**Summary Of The Review:**

It is challenging to evaluate the soundness of the proposed approach due to the clarity and quality of the paper. Furthermore, the proposed baseline values are not motivated properly. As a result, I believe this paper requires a major revision.

---

### Official Review · Reviewer_y3yp · 2022-10-26

**Confidence:** 2
**Correctness:** 2
**Technical Novelty And Significance:** 2
**Empirical Novelty And Significance:** 2
**Recommendation:** 3

**Clarity, Quality, Novelty And Reproducibility:**

 I am not an expert in this filed. However, as a reader, this work is unreadable and requires a lot of prior information. Besides, there are a lot of undefined notations or terms.

**Strength And Weaknesses:**

This method combines Shapley values with the Integrated Gradients method for better accuracy of locating relevant features while causing a trade-off to the efficiency of the original Integrated Gradients method. The novelty of such a combination is limited. The sampling strategy to reduce computation of SHAPLEY VALUE is also not novel.  Moreover, it's not clear how to set baseline values when calculating v(S) in Algorithm 1. It's also not clear how to directly set Di as baseline values in Algorithm 2. Why the authors use Integrated Gradients is also questionable. Each player's contribution is also calculated in the algorithm1. Why is the Integrated Gradients method necessary to calculate the contribution? The overall writing of the methodology is confusing.

Moreover, there are a lot of notations confusing readers: what is v in Eq (1)? What is the difference between S1 and s1?

**Summary Of The Paper:**

This work focuses on the task of attributing the prediction of DNN to its input features and try to address the baseline issues. The work proposes to find a set of baseline values corresponding to Shapley values.To solve the computation dilemma of Shapley value,  a proportional sampling method (Effective Shapley value, ES) is proposed to well simulate the ratios between the Shapley values
of features. Besides, the submition proposed Shapley Integrated Gradients (SIG) to combine Integrated Gradients with ES, to achieve a good balance between efficiency and effectiveness. I am not an expert in this filed. However, as a reader, this work is unreadable and requires a lot of prior information. Besides, there are a lot of undefined notations or terms.

**Summary Of The Review:**

I cannot recommend accepting this work. Although I am no working in this field, this work is far from a ready work for publish.

---

### Official Review · Reviewer_QQsg · 2022-11-02

**Confidence:** 5
**Correctness:** 2
**Technical Novelty And Significance:** 2
**Empirical Novelty And Significance:** Not applicable
**Recommendation:** 3

**Clarity, Quality, Novelty And Reproducibility:**

As discussed in the weaknesses section there is a lack of clarity which limits reproducibility. On top of that the paper completely ignores a whole body of work critically evaluating interpretability methods. There is no evidence provided that the proposed approach is actually working reliably.

**Strength And Weaknesses:**

Weaknesses:
- Experimental validation
  -  Validation in Fig 2.  The entire experimental setup is not provided. It is not possible to actually implement this experiment and it is not possible to verify why it would make sense. Additional detail is needed.
 - Figures 3, 4, 5: it is not obvious at all why these visualisations are better than any other interpretability method. Why is this method showing the right result?
 - Positioning the work relative to work looking at the limits and potential issues with interpretability methods. There is not discussion whether this method performs well w.r.t. to any of these papers discussing the effectiveness of interpretability methods and their potential limitations.
    - e.g. the evaluation by Hooker et al. A benchmark for interpretability methods in deep neural networks
    - Adebayo et al. Sanity checks for saliency maps
    - Kindermans et al. The (un) reliability of saliency methods,
   - Wilming et al. Scrutinizing XAI using linear ground-truth data with suppressor variables

- Clarity
  - On page 3 the shapely value is defined properly. However It is not clear how the Aumann Shapley value should be interpreted with a neural network. The citation for the Aumann Shapley value is not available in open access nor is it discussed properly on websites such as wikipedia. With the limited discussion here I am not able to properly understand the nuances of this. For example it is not clear how the players in the game map to an image classification problem.
  - On Page 3 at the bottom: We make the hypothesis that baseline sample x′ is considered as background in game and all points alongside P2 are considered as complete set I in game. It is never defined what background in the game actually means. How this should be interpreted.
  - Theorem 1: please provide an intuition in the main paper. There is also no sufficiently formal definition of all components in the theorem to actually understand the proof in the supplementary.
  - Page 5: partial coalition appears twice in the document and is not defined. It is not clear from the context what it actually is.
Strengths:
 - Algorithm 1 and 2: make sure everything is defined properly. It is not clear for example how you can sample according to equation 1 as mentioned in algorithm 1. What is shown there is a sum, not a distribution.
  - "We first set players coalition as a single player, like 80*80 area as a player in image." This sentence is incredibly confusing. How do you change between a player contribution or not.

Strengths:
N/A

**Summary Of The Paper:**

The paper proposes ideas to combine integrated gradients and shapely values effectively. The problem according to the paper is that shapely values  are too expensive to be computed directly. The paper wants to use the ideas from shapely values to find a proper baseline for integrated gradients. The paper proposes effective shapely value, an approximation of shapely values that can be computed more effectively.

The paper claims that experimental results show that a combination of shapely values with integrated gradients provides good results.

**Summary Of The Review:**

The paper does not provide sufficient experimental justification of the proposed approach to be accepted. It ignores a lot of issues with interpretability methods and their evaluation that are discussed in literature.

---

### Decision · Program_Chairs · 2023-01-20

**Decision:**

Reject

**Justification For Why Not Higher Score:**

- Main claims not substantiated
- Exposition needs a major revision
- No rebuttal

**Justification For Why Not Lower Score:**

N/A

**Metareview: Summary, Strengths And Weaknesses:**

Authors continue the line of work of improving the understanding of DNNs by attributing predictions to the input features. In particular, the authors propose Effective Shapley value (ES), a proportional sampling method to estimate the ratios between Shapley values. These are then combined with existing Integrated Gradients method to enable a tradeoff between efficiency and effectiveness. Empirical study seems to indicate that the proposed method improves upon common baseline values while maintaining similar computational costs.

The reviewers pointed out several weaknesses, primarily with respect to the clarity of exposition and suggested a plethora of improvements, including concrete detailed changes to the exposition. In the current state, the main contributions are not clear and the main claims are not sufficiently substantiated. The authors didn't provide any rebuttal, and I will hence suggest the rejection of the manuscript.